# In-Depth Analysis of the *N*-Glycome of Colorectal Cancer Cell Lines

**DOI:** 10.3390/ijms24054842

**Published:** 2023-03-02

**Authors:** Di Wang, Valeriia Kuzyk, Katarina Madunić, Tao Zhang, Oleg A. Mayboroda, Manfred Wuhrer, Guinevere S. M. Lageveen-Kammeijer

**Affiliations:** 1Center for Proteomics and Metabolomics, Leiden University Medical Center, Postbus 9600, 2300 RC Leiden, The Netherlands; 2Division of Bioanalytical Chemistry, Vrije Universiteit Amsterdam, 1081 HV Amsterdam, The Netherlands; 3Department of Cellular and Molecular Medicine, Copenhagen Center for Glycomics, University of Copenhagen, 2200 Copenhagen, Denmark; 4Analytical Biochemistry, Groningen Research Institute of Pharmacy, University of Groningen, 9700 AD Groningen, The Netherlands

**Keywords:** *N*-glycosylation, colorectal cancer, cell line, glycosyltransferases, transcription factor, porous graphitized carbon liquid chromatography, mass spectrometry

## Abstract

Colorectal cancer (CRC) is the third most commonly diagnosed cancer and the second leading cause of cancer deaths worldwide. A well-known hallmark of cancer is altered glycosylation. Analyzing the *N*-glycosylation of CRC cell lines may provide potential therapeutic or diagnostic targets. In this study, an in-depth *N*-glycomic analysis of 25 CRC cell lines was conducted using porous graphitized carbon nano-liquid chromatography coupled to electrospray ionization mass spectrometry. This method allows for the separation of isomers and performs structural characterization, revealing profound *N*-glycomic diversity among the studied CRC cell lines with the elucidation of a number of 139 *N*-glycans. A high degree of similarity between the two *N*-glycan datasets measured on the two different platforms (porous graphitized carbon nano-liquid chromatography electrospray ionization tandem mass spectrometry (PGC-nano-LC-ESI-MS) and matrix-assisted laser desorption/ionization time of flight-mass spectrometry (MALDI-TOF-MS)) was discovered. Furthermore, we studied the associations between glycosylation features, glycosyltransferases (GTs), and transcription factors (TFs). While no significant correlations between the glycosylation features and GTs were found, the association between TF CDX1 and (s)Le antigen expression and relevant GTs FUT3/6 suggests that CDX1 contributes to the expression of the (s)Le antigen through the regulation of FUT3/6. Our study provides a comprehensive characterization of the *N*-glycome of CRC cell lines, which may contribute to the future discovery of novel glyco-biomarkers of CRC.

## 1. Introduction

Colorectal cancer (CRC) is the third most commonly diagnosed cancer, accounting for 10.0% of all cancer cases, and is the second leading cause of cancer death, accounting for 9.4% of all cancer deaths worldwide [1]. Advances in diagnostics, particularly in population screening, and therapeutic treatments over the past 50 years, have led to significant reductions in incidence and mortality [2,3]. However, many tumors are still only detected at advanced stages which often results in treatment failure [4].

Glycosylation is a well-known hallmark of cancer [5] and, therefore, investigating the expression and regulation of glycosylation in relation to CRC is essential for the discovery of potential cancer-associated biomarkers. The complex, nontemplate-based biosynthetic pathways of glycosylation create an enormous structural diversity and a rich pool of putative biomarkers. Furthermore, the discovery of glycomic biomarkers can be derived from various data arrays. For example, one may investigate the entire glycomic pattern of a biofluid (e.g., salivary glycome for oral cancer) [6] or tissue sample (colonic tissues for CRC) [7], or focus on glycan signatures of a specific protein group (such as immunoglobulins in the serum of ovarian cancer patients) [8]. Alternatively, the search can be narrowed down to the glycosylation of one glycoprotein that is known to be cancer-associated, as it was probed for the glycosylation of haptoglobin in hepatocellular carcinoma and carcinoembryonic antigen in CRC [9,10,11].

To model the response to treatment and biomolecular signatures, cell lines are extensively used in cancer research thereby being capable of revealing potential biomarkers [12,13]. The established cell lines are well characterized at the gene and protein expression levels and retain tissue molecular features with regard to DNA, RNA, and proteins making them an essential tool in preclinical drug screening experiments [14]. An integrated data analysis approach, which combines DNA mutations, RNA, and protein expression levels, has been used to classify the CRC cell lines into two distinct groups: colon-like cell lines with a characteristic expression of gastrointestinal differentiation markers and undifferentiated cell lines that show overexpression of both epithelial-mesenchymal transition (EMT) pathway and transforming growth factor β (TGF-β) signaling [15]. However, the glycan signatures are often neglected in these studies, partly due to their high heterogeneity and complexity. Previous research found a high diversity in the *N*-glycomic fingerprints between different CRC cell lines analyzed with matrix-assisted laser desorption/ionization time of flight-mass spectrometry (MALDI-TOF-MS), which can be used as glycobiological tumor model systems [16].

It is important to note that seemingly small differences in glycan structures may result in significant physiological outcomes. For instance, a shift from dominating α2-3 sialylation in sialic acid linkage isomers towards α2-6 sialylated species has been reported to be indicative of early ovarian cancer [17] and was found to accompany drug resistance in metastatic CRC [18]. Our group has previously explored the glycosylation of glycosphingolipids (GSLs), *N*- and *O*-glycans of CRC cell lines [16,19,20], and CRC tissues [21]. Colon-like cell lines exhibited a high abundance of sialyl Lewis^A/X^ (sLe^A/X^) antigens on *O*-glycans and GSLs glycans, whereas undifferentiated cell lines presented an upregulation of blood group antigens on GSLs glycans and truncated α2-6 core sialylated glycans on *O*-glycans [19,20]. Yet, the previously performed *N*-glycomic analysis of CRC cell lines measured with MALDI-TOF-MS had limitations in resolving isomers, as no separation dimension was applied, and, from the glycan isomers perspective, only differences in sialic acid linkage could be resolved via linkage-specific sialic acid derivatization [16]. Certain glycan motifs that are defining pathological processes (such as sLe^A/X^ glycan antigens) are difficult to identify reliably without an in-depth structural analysis. All these considerations prompt the need for a more detailed structural analysis of CRC cell lines’ glycosylation.

Glycosylation is a dynamic process in response to microenvironment demands and is only partially dependent on glycosyltransferase expression. The localization of these enzymes, the enzymes synthesizing the monosaccharide precursors, the availability of nucleotide sugar donors and transporters, and alterations of the peptide backbone substrate all contribute to the final glycan signature [22,23,24,25]. In this study, to gain a better understanding of the underlying mechanisms related to newly found glycosylation alterations, we combined mRNA expression data of CRC cell lines with its *N*-glycomic profile.

With these considerations, we have characterized and provided insights into the structural diversity of the *N*-glycome and investigated the regulation of the *N*-glycan expression in 25 CRC cell lines. In total, a number of 139 *N*-glycans were structurally identified with a porous graphitized carbon nano-liquid chromatography electrospray ionization tandem mass spectrometry (PGC-nano-LC-ESI-MS/MS). This method provides the in-depth structural characterization of isomeric species due to the separation of (isomeric) *N*-glycan species. Moreover, we investigated the association of glycomic features and relevant glycosyltransferases (GTs) in CRC cell lines. Overall, our findings enrich the current understanding of CRC cell lines as model objects and pave the way for discovering potential cancer-associated targets for the treatment of CRC.

## 2. Results

### 2.1. Diversity of CRC Cell Line N-Glycome

To characterize the *N*-glycan profiles of 25 CRC cell lines, released *N*-glycans were measured on a PGC-nano-LC-ESI-MS/MS system in negative mode, providing fragment-specific diagnostic ions derived from cross-ring and glycosidic-linkage fragmentation to characterize the glycans and differentiate isomers [26,27,28,29,30,31,32]. In total, 139 different *N*-glycans were structurally confirmed ranging in size from 4 up to 14 monosaccharides per structure (Appendix A). Relative abundances of the *N*-glycans, based on the first three isotopic peaks of singly and doubly charged ions, were determined, which may somewhat deviate from the actual relative abundances due to possible biases in sample preparation and mass spectrometric detection. Our analysis revealed a high degree of diversity in *N*-glycomic profiles among the studied CRC cell lines (Figure 1 and Appendix A). The undifferentiated Caco2 cell line expressed high phosphorylated oligomannosidic type structures and complex *N*-glycans with Lewis antigens, while the colon-like SW1463 cell line showed a higher diversity of sialylated *N*-glycans (Figure 1). The undifferentiated cell line HCT116 expressed a higher diversity of *N*-glycans compared to the colon-like cell line HT29 and, apart from typically predominant (phosphorylated) oligomannosidic species (Appendix A), was further characterized by the sialylated complex and hybrid *N*-glycans (Appendix A). Notably, LacdiNAc (GalNAcβ1-4GlcNAc) motifs were observed for two isomers with composition H4N5S1 with a relative abundance of 0.7% and 0.4% (Appendix A). While the colon-like cell line HT29 was characterized by a high abundance of oligomannosidic *N*-glycans, a high abundance of paucimannosidic types (3.5%), along with a sulfated *N*-glycan (H6N3Su1) with a relative abundance of 0.8% (Appendix A), which is in contrast to HCT116. Structural information of the identified *N*-glycans can be found in Appendix A.

The *N*-glycomic phenotypes of CRC cell lines were relatively quantified and assigned into different glycosylation groups based on specific glycan types (paucimannosidic, oligomannosidic, complex, and hybrid), glycan epitopes (core/antennae fucosylation, sialylation, LacdiNAc motifs, Lewis type antigens, and H blood group antigen), as well as additional modifications (sulfation and phosphorylation) (Appendix A). Principal component analysis (PCA) of the abundance of *N*-glycan derived traits did not reveal a clear grouping between the glycosylation features and the CRC cell line differentiation (Appendix A). For instance, colon-like cell lines LS174T and LS180, as well as the undifferentiated cell line SW48 and the unassigned cell line SW1398, clustered together at the top of the score plot due to the high abundance of paucimannosidic *N*-glycans, (core/antenna) fucosylation, and Le^A/X^ as well as sLe^A/X^ antigens (Appendix A). At the right-bottom panel, most colon-like cell lines clustered together with a few undifferentiated cell lines, including SW620, Caco2, RKO, and HCT115, and the unassigned cell line HCT8, driven by the high expression of oligomannosidic *N*-glycans, phosphorylation, bisection, Le^B/Y^ antigens, and H antigens (Appendix A).

### 2.2. CRC Cell Line Classes Show Different Average N-Glycosylation Features

We further explored the distribution of *N*-glycosylation features in two different classifications of CRC cell lines (classification was based on gene expression) [15]. Briefly, colon-like cell lines (Colo205, HT29, SW1116, KM12, SW948, LS174T, SW1463, LS180, and WiDr) were characterized by the high expression of gastrointestinal differentiation markers, while the undifferentiated cell lines (Caco2, Co115, LOVO, DLD-1, HCT15, HCT116, RKO, SW48, SW480, and SW620) were marked by upregulation of the EMT and TGF-β signatures. Cell lines T84, Colo320, LS411N, SW1398, C10, and HCT8, which were not characterized for their differentiation status, were kept as unassigned.

The profound diversity of *N*-glycomic profiles was revealed among CRC cell lines (Figure 2). Regarding Lewis antigens, Le^B/Y^ were only detected in five specimens (colon-like cell lines HT29, SW948, and LS180, and the undifferentiated cell lines SW480 and HCT8) with the highest abundance in HCT8 (0.8%), (Figure 2 and Appendix A). The LS180 cell line expressed the highest abundance of Le^A/X^ antigens (5.5%), followed by 4.5% in the colon-like cell line LS174T. Only three lines were found to express sLe^A/X^ antigens (LS174T, LS180, and SW1116), all of them being colon-like cell lines (Figure 2 and Appendix A). H blood group antigen was most abundant in the unassigned cell line HCT8 (2.3%), followed by the two undifferentiated cell lines Caco2 (1.0%) and SW480 (0.7%). In regard to sialylation, α2-3 sialylation was found to be highest in the colon-like cell line SW1463 (19.8%), followed by the undifferentiated cell lines Co115 (17.0%) and DLD-1 (14.9%). To note, α2-6 sialylation was expressed in all CRC cell lines, with the highest expression in the undifferentiated cell line HCT116 (25.4%). Furthermore, paucimannosidic *N*-glycans were detected in all cell lines, with the highest expression in the undifferentiated cell line SW48 (15.1%). Meanwhile, SW48 expressed the second most abundance of triantennary *N*-glycans (1.1%). The highest abundance of this feature, of all cell lines, was found in the unassigned cell line C10 (1.7%). Tetraantennary *N*-glycans were only expressed in three cell lines, including the unassigned cell line SW1398 (0.9%) and the undifferentiated cell lines SW48 and LOVO (Figure 2 and Appendix A). The distribution of glycosylation features based on cell line classifications is illustrated in Appendix A. The *N*-glycans carrying LacdiNAc with or without fucose linked to GlcNAc via α1-3 linkage (GalNAc(β1-4)[+/−Fuc(α1-3)]GlcNAc(β1-3)GalNAc(β1-4)[+/−Fuc(α1-3)]GlcNAc(β1-), phosphorylation, bisection, and α2-3 sialylation, as well as the paucimannosidic *N*-glycans, were highly expressed in undifferentiated cell lines with a low expression of antenna fucosylation, Le^B/Y^, Le^A/X^, and the hybrid *N*-glycans (Appendix A). In contrast, the colon-like cell lines demonstrated a high abundance of sulfation, Le^A/X^, sLe^A/X^, Le^B/Y^, antenna fucosylation, and high levels of oligomannosidic, as well as hybrid *N*-glycans. Notably, antenna fucosylation and Le^A/X^ were found to be expressed significantly higher in colon-like cell lines in comparison with undifferentiated cell lines (Appendix A).

### 2.3. CRC Cell Line N-Glycosylation Analysis by PGC-Nano-LC-ESI-MS and MALDI-TOF-MS Provides Complementary Information

Previously, we performed a total *N*-glycome characterization of the same CRC cell lines panel using MALDI-TOF-MS [16]. A Spearman correlation analysis was conducted to determine the similarity between the two *N*-glycan datasets measured on two different platforms (PGC-nano-LC-ESI-MS and MALDI-TOF-MS) (Figure 3, Appendix A). Significant correlations were found for the relative abundance of paucimannosidic and hybrid *N*-glycans, however, no correlation between the datasets for oligomannosidic and complex *N*-glycans was observed. The HexNAc ≥ Hex glycosylation trait (MALDI-TOF-MS) correlated positively with the bisection trait (PGC-nano-LC-ESI-MS). The HexNAc ≥ Hex trait can be seen as an indicator of bisection in complex-type *N*-glycans, though other glycosylation features such as truncated antennae, LacdiNAc motifs, Sd^a^ antigens, and blood-type A motifs can also shift the composition towards more HexNAc units, making the assessment of bisection levels ambiguous. In the current study, the presence of bisection is confirmed by negative mode MS/MS fragmentation spectra as well as retention time on PGC, thereby attributing the bisection trait with high confidence. As for antennarity, the diantennary *N*-glycans correlated between the two datasets. Moreover, (core)fucosylation in the present study correlated significantly with (mono)fucosylation measured by MALDI-TOF-MS. Antennae fucosylation (PGC-nano-LC-ESI-MS) showed a significant and strong correlation with multifucosylation (MALDI-TOF-MS), which also correlated positively with Le^A/X^, sLe^A/X^, Le^B/Y^, and H antigens measured in the present study. With the current application, the location of fucose can be identified with more certainty through diagnostic MS/MS fragments and the absence of fucose migration [33]. To illustrate the presence of the Z ions, *m*/*z* 350 and 553, as well as the Y ions, *m*/*z* 368 and 571, which indicates the fucose being linked to the innermost GlcNAc of *N*-glycans, i.e., core fucosylation. The presence of terminal B ion cleavages, *m*/*z* 510, and C ion, *m*/*z* 528, indicates terminally linked fucose on the *N*-glycan antenna commonly associated with Lewis antigen structures [26]. A strong positive correlation was found for (α2-6) sialylation in both datasets. However, no significant correlation was observed for α2-3 sialylation.

Next to features overlapping with the MALDI-TOF-MS results, the PGC method provided additional, unique insights into CRC cell line *N*-glycosylation. The additional separation of isomers using the PGC-nano-LC-ESI-MS platform alongside diagnostic ions produced in the MS/MS spectra provided an in-depth structural characterization of the *N*-glycans and allowed us to identify the sequence and location of monosaccharides such as core fucosylation, Lewis structures, and blood group motifs. The presence of the D-221 ion, which is formed from the additional loss of the β1-4 linked GlcNAc from the D ion of bisected *N*-glycans, indicated the presence of bisecting GlcNAc, ^1,3^A cleavage ions, indicating the composition of antennae [26]. MS/MS spectra of the selected *N*-glycans were presented in the Appendix A. On the other hand, while present in low abundance, the MALDI-TOF-MS dataset has revealed more high-branching complex and hybrid structures. To summarize, next to the already obtained MALDI-TOF-MS information, the usage of PGC-nano-LC-ESI-MS proved to be a comprehensive and complementary tool for the structural identification of *N*-glycans.

### 2.4. The Correlation of N-Glycosylation Features and Relevant GTs in CRC and AML Cell Lines

Recently, we explored the *N*-glycome of various acute myeloid leukemia (AML) cell lines and revealed the diversity of their *N*-glycosylation [34]. In the current study, we investigated the correlation of *N*-glycosylation features and corresponding GTs for both the AML and CRC cell lines (Figure 4A, Appendix A). The transcriptomic of the GT’s expression in the AML cell lines was obtained from a publicly available dataset [35]. Overall, a significant association was found for (core)fucosylation with FUT8 for the CRC cell lines. However, no significant correlation was observed for the AML cell lines (Figure 4A). The expression of (core)fucosylation between the AML and CRC cell lines was further explored, and this revealed a higher expression of (core)fucosylation in the AML cell lines (Figure 4B), although no significant difference was found between the AML and CRC cell lines regarding the transcriptomic expression level of FUT8 (Appendix A). FUT4/7/9 are involved in the expression of sLe^X^ [36,37]. Contrary to most CRC cell lines, significant correlations were observed between Le^A/X^, sLe^A/X^, and FUT4/7/9 for the AML cell lines (Figure 4A). An overall higher expression of Le^A/X^ and sLe^A/X^ was found for the AML cell lines in comparison to the CRC cell lines (Figure 4B), with accordingly higher expression of the FUT4/7 in the AML cell lines compared to the CRC cell lines (Appendix A). The AML cell lines exhibited a high degree of (α2-3/6) sialylation, relatively, compared to the CRC cell lines (Figure 4B), which might result from the increased transcriptomic expression of ST3GAL1/3/4/6 and ST6GAL1 in AML cell lines in comparison to the CRC cell lines (Appendix A).

### 2.5. The Correlations of TFs with N-Glycosylation Features and Corresponding GTs in CRC Cell Lines

To gain insight into the regulation of *N*-glycosylation in the CRC cell lines, the transcriptomic expression of relevant genes (GTs and certain transcription factors (TFs)) was selected and obtained from the Cancer Cell Line Encyclopedia public dataset based on their involvement in the biosynthesis of *N*-glycans [15]. A previous study has shown that certain TFs, such as TFs CDX1, ETS2, HNF1A, HNF4A, MECOM, and MYB are notably expressed in colon-like cell lines. In contrast, undifferentiated cell lines showed a significantly higher expression of other TFs (e.g., MLLT10, MSX1, SIX4, ZNF286A, and ZNF286B) [19]. In this study we found the GANAB gene (encoding the GT responsible for removal of the two innermost α1-3 linked glucose residues from Glc(2)Man(9)GlcNAc(2) oligosaccharide precursor) [38] to correlate significantly with ETS2. This, in turn, was positively associated with the oligomannosidic trait, although the latter relationship was insignificant (Figure 5A; Appendix A). A significant positive association was demonstrated between MGAT1 (encoding a GT essential for the conversion of oligomannosidic to hybrid type *N*-glycans) and HNF4A, which showed a positive trend towards the association with hybrid type *N*-glycans expression (not significant). Additionally, MGAT3 (GT catalyzing the addition of *N*-acetylglucosamine in β1-4 linkage to core mannose of *N*-glycan to form bisected *N*-glycans) was significantly associated with HNF1A and HNF4A which, subsequently, coincreased along with the *N*-glycans bisection (Figure 5A). The enzyme involved in the biosynthesis of tri- and tetraantennary *N*-glycans (MGAT4B) positively associated with the TFs ETS2, HNF1A, HNF4A, MECOM, and MYB which, however, exhibited negative correlations (not significant) with tri- and tetraantennary *N*-glycans. In our study, FUT3, involved in the biosynthesis of Le^A/X/B/Y^ antigens [39,40,41], showed a positive association with ETS2 and MYB, which significantly correlated with the Le^A/X^ antigens, as well as with CDX1, which was in positive correlation with sLe^A/X^ glycosylation. A similar pattern was observed for FUT4, involved in the expression of (s)Le^X^ [36,42,43]. Notably, FUT6, its corresponding GT involved in the biosynthesis of E-selectin ligand sLe^X^ [44,45], positively correlated with CDX1 and also revealed a positive association with the sLe^A/X^ glycosylation trait. FUT9, a GT catalyzing the biosynthesis of the Le^X^ antigen, positively correlated with HNF4A and showed a trend toward positive correlation with Le^A/X^ (Figure 5B). Moreover, FUT2, catalyzing the biosynthesis of terminal α1-2 fucose in the H blood group antigen, showed a positive correlation with ETS2. However, no correlation was found between this TF and the H blood group antigen glycosylation trait, however, a slight positive correlation was seen with Le^B/Y^ (Figure 5B).

## 3. Discussion

In this study, an in-depth characterization of the *N*-glycome of 25 CRC cell lines was conducted using PGC-nano-LC-ESI-MS/MS and revealed the diversity of the *N*-glycomic signatures among the CRC cell lines. Our platform allowed the separation of isomeric *N*-glycan species with structural elucidation based on MS/MS fragmentation patterns in negative ion mode.

Currently, carcinoembryonic antigen (CEA) and carbohydrate antigen (CA)19-9 are used as tumor markers in clinical practice for the diagnosis, prognosis, and monitoring of CRC patients [46,47,48,49,50]. Nonetheless, as these markers exhibit low specificities and suboptimal sensitivities, especially in the early stages of the disease [51,52,53,54], there is an urgent need for novel biomarkers and therapeutic targets. Cancer progression frequently correlates with *N*-glycome alterations [55], making the discovery of tumor-associated carbohydrate antigens (TACAs) a promising area for identifying the new molecular signatures of tumors for improved diagnostics, stratification, and targeted treatment [21,56]. Recently, we have investigated the glycomic profiles of GSL glycans in relation to CRC and revealed that colon-like cells are dominated by a high expression of glycans carrying (s)Le antigens, while, undifferentiated cell lines showed an increased level of glycans with the terminal blood group antigens H, A, and B [20]. When it comes to *O*-glycans, colon-like cell lines presented an upregulation of (s)Le antigens, while undifferentiated cell lines were dominated with truncated α2-6 core sialylated *O*-glycans, with specific cell lines expressing high levels of H blood group antigen [19]. Consistent with these previous findings, the present study demonstrated a high expression of (s)Le antigens in CRC colon-like cell lines (Figure 2). However, H blood group antigens were only found in limited quantities in undifferentiated cell lines. In another study, focusing on the comprehensive TACAson *O*-glycans in CRC primary tissues, revealed that specific (s)Le core two were exclusively expressed in tumors and absent in the normal mucosa from the same patients [21]. Additionally, *N*-glycan expression analyzed in the same set of tumor samples, revealed several *N*-glycans carrying (s)Le antigens exclusively expressed in tumor samples [57]. In this study, we found that (s)Le antigens were highly expressed in the colon-like cell lines LS180 and LS174T (Appendix A). A decrease of core fucosylated diantennary *N*-glycans and an increase of highly galactosylated, highly sialylated, and tetraantennary *N*-glycans were found in the plasma of CRC patients, in comparison to healthy individuals [58]. That variety of glycosylation phenotypes originating from different studies only corroborates the complexity of the glycosylation machinery and its multilevel regulation. Therefore, an integration of the glycosylation features on the glycoprotein and glycolipid levels appears as the most fruitful biomarker discovery strategy for further studies.

It is important to note that the employed method of analysis has a significant impact on the resulting data pool. Previously, we characterized the total cell *N*-glycome of these CRC cell lines by MALDI-TOF-MS and concluded that they have a high potential to be used as glycobiological tumor model systems [16]. However, while MALDI-TOF-MS is efficient for high-throughput glycan profiling, it lacks the ability to separate glycan isomers or provide structural information when no separation is applied prior to the measurement [26]. Moreover, a derivatization strategy needs to be performed to prevent the loss of sialic acids during laser-assisted ionization [59]. In this study, we took a deeper dive into the total *N*-glycome of the CRC cell line panel by structural analysis using PGC-nano-LC-ESI-MS/MS, which is capable of separating isomeric glycans [26]. A total of 139 *N*-glycans could be identified and structurally characterized (Appendix A). Statistically significant correlations were observed between the glycosylation features of both datasets (MALDI-TOF-MS versus PGC-nano-LC-ESI-MS; Figure 3 and Appendix A). However, some differences in the resulting data were also observed between the two methods. While PGC-nano-LC-ESI-MS provided a better insight into the midrange mass *N*-glycan diversity, the coverage of the high-mass *N*-glycans was rather limited compared to MALDI-TOF-MS. This could be explained by the dynamic range and/or the sensitivity of the two methods. The relative intensities of these structures in the MALDI-TOF-MS dataset are relatively low and, as no separation is occurring in this method, all the present structural isomers’ mass contributes to the total intensity. In the case of PGC, the isomers would be separated and the intensity per isomer would be inevitably lower.

In order to further explore this relationship, we have performed a correlation analysis between *N*-glycomic profile traits and TFs and sought unique gene and *N*-glycan expression patterns in the CRC cell lines compared to their AML counterparts (Figure 4). These two datasets revealed a highly different expression of glycosylation features accompanied by an alternative activation of GTs and regulation systems. Increasing evidence demonstrates *N*-glycans influencing the processing and functioning of GTs, including their secretion, stability, and substrate–acceptor affinity [60,61]. Growing evidence suggests that *N*-glycans play a crucial role in regulating the processing and function of GTs through facilitating the folding of the polypeptide chain, ensuring the correct subcellular localization of the protein, and preventing protein aggregation [62]. Thus, the observation of higher expression of (core)fucosylation, (α2-3/6) sialylation, Le^A/X^, and sLe^A/X^ in AML cell lines compared to the CRC cell lines may partially explain the differences in the GT expression observed between the two types of cancer (Figure 4B). Furthermore, it is possible that TFs also play a role in regulating GT expression. However, little is known about TFs that can directly participate in the regulation of the expression of GTs. Further research is needed to fully understand the mechanisms by which TFs regulate glycans in cancer and how this knowledge can inform targeted treatment strategies for CRC.

The biosynthesis and expression of *N*-glycans are primarily attributed to the series of actions of GTs and glycosidases. Hitherto, little was explored for potential regulatory layers. A separate study demonstrated that several regulatory processes such as post-transcriptional, translational, and protein degradation regulation are also involved in controlling steady-state protein abundances after the production of mRNA [63]. This means that transcript levels may be insufficient to accurately predict protein expression levels [64]. With regard to the expression of GTs, a previous study reported that signal peptide peptidase-like 3 (SPPL3) alters cellular *N*-glycosylation by the proteolytic release of the ectodomain of various GTs and glycosidase such as *N*-acetylglucosaminyltransferase V, β1-3, *N*-acetylglucosaminyltransferase 1 and β1-4 galactosyltransferase 1 [65]. Subsequently, the higher expression of SPPL3 leads to hypoglycosylation, and decreased SPPL3 expression causes hyperglycosylation [65]. Meanwhile, the activity of B3GNT5, a key enzyme responsible for the synthesis of the neolacto-series of GSLs, was suppressed by SPPL3, which notably affects the expression of GSL glycans on the cell surface [66]. Another study demonstrated that protease β-site amyloid precursor protein-cleaving enzyme 1 (BACE1) is responsible for the proteolytic cleavage of the beta-galactoside alpha-2,6-sialyltransferase 1 enzyme (encoded by ST6GAL1) [67]. These findings suggest that the protease-mediated degradation of GTs may result in a poor correlation between transcript level and protein level. That may partly explain the differences in correlations between mRNA expression of GTs with glycosylation features in CRC and AML found in the present study (Figure 4). In addition, *N*-glycan diversity may be influenced by several factors, such as substrate availability for GTs and glycosidases in Golgi, nucleotide sugar metabolism, transport rates of the glycoprotein through the lumen of the ER and Golgi, and the proximity of an *N*-glycan attachment sequon to a transmembrane domain [68], which may also contribute to the unexpected observations between GTs and the glycosylation features in CRC cell lines.

Another relationship we have explored is an association between highly fucosylated *N*-glycans and the expression of CDX1—caudal-related homeobox protein 1 (CDX1)—that, as a TF, plays an essential part in the development, differentiation, and homeostasis of the gut [69,70], and this link was previously reported for CRC cell lines [16]. However, in the present study, no significant correlation was revealed, which may be attributed to the application of correlation analysis on all CRC cell lines instead of the CRC cell lines with a high expression of CDX1. However, (s)Le structures, which are promising targets for novel treatment strategies [21], had a strong correlation with CDX1, which, in turn, is positively associated with FUT3/6 (Figure 5). Corroborating our findings, a previous study showed that a high level of CDX1 expression and less invasive and aggressive phenotype was associated with a higher abundance of multifucosylation on *N*-glycans in the CRC cell lines and was supported by the upregulation of GTs involved in antenna fucosylation such as FUT3/5/6 (Figure 5) [16,71], Taking into account the previous and current findings, we hypothesize that CDX1 may play an essential role in the formation of (s)Le antigens on colon-like cell lines via the regulation of the corresponding GTs (mainly FUT3/6). However, further research is needed to validate if other TFs may conduct the fucosylation patterns in the undifferentiated cell lines (e.g., Caco2).

In conclusion, our detailed analysis of 25 CRC cell lines revealed a distinct diversity of *N*-glycomic profiles and showed a strong relationship with previous findings for the same set of cell lines measured by MALDI-TOF-MS. Our results suggest that, from the glycosylation features’ point of view, using different platforms for similar samples does not yield conflicting results and demonstrates a high degree of similarity. Instead, the observed differences are complimentary in nature. Our data also indicate that certain glycosylation features have a cell type-specific distribution among different CRC cell line classifications. Namely, colon-like cell lines exhibit a relatively high expression of (s)Le antigens, while undifferentiated cell lines are characterized by an upregulation of paucimannosidic, bisected *N*-glycans, α2-3 sialylation, and *N*-glycans carrying with (fucosylated) LacdiNAc. Associations were observed between TF CDX1 and (s)Le antigen, and corresponding GTs indicated that the potential mechanism of expression and regulation of glycosylation features, especially, (s)Le antigen, which might be under the control of the corresponding GTs FUT3/6 regulated by CDX1.

## 4. Material and Methods

### 4.1. Materials

Trifluoroacetic acid (TFA), guanidine hydrochloride (GuHCl), sodium borohydride, sodium chloride, DL-dithiothreitol (DTT), ammonium bicarbonate (ABC), fetuin from fetal bovine serum, cation exchange resin Dowex 50W X8 and ammonium acetate were obtained from Sigma Aldrich (St. Louis, MO, USA). Ethanol, NaCl, and methanol (MeOH) were purchased from Merck (Darmstadt, Germany). Acetonitrile LC-MS grade (MeCN) was acquired from Biosolve (Valkenswaard, The Netherlands), KOH and glacial acetic acid were obtained from Honeywell Fluka (Charlotte, NC, USA), and PNGase F (Flavobacterium meningosepticum recombinant in *E. coli*) from Roche (Mannheim, Germany). SPE bulk sorbent Carbograph was from BGB Analytik USA LLC (Alexandria, VA, USA), MultiScreen^®^ HTS 96-multiwell plates (pore size 0.45 m) with a high protein-binding membrane (hydrophobic Immobilon-P PVDF membrane) and 96-well PP Microplate were obtained from Millipore (Amsterdam, The Netherlands). The 96-well PP filter plate from Orochem technologies (Naperville, IL, USA) and isopropanol were purchased from Biosolve Chimie (Dieuze, France). T75 cell culture flasks were obtained from Greiner-Bio One B.V. (Alphen aan de Rijn, The Netherlands) and Hepes-buffered RPMI 1640 and Dulbecco’s Modified Eagle (DMEM) culture media were obtained from Gibco (Paisley, UK). Fetal bovine serum (FBS) and penicillin/streptomycin were bought from Invitrogen (Carlsbad, CA, USA) and 0.5% trypsin-EDTA solution 10× was from Santa Cruz Biotechnology (Dallas, TX, USA). Ultrapure water (mQ) generated by the ELGA system (ELGA, High Wycombe, UK) maintained at ≥18 MΩ was used for all solvent preparations and washing steps.

### 4.2. Cell Lines and Cell Culture

Human CRC cell lines were provided by the Department of Surgery of the Leiden University Medical Center (LUMC, Leiden, the Netherlands) and the Department of Pathology of the VU University Medical Center (VUmc, Amsterdam, The Netherlands). The LUMC cell lines were cultured with Hepes-buffered RPMI 1640 medium with 2 mM L-glutamine with the supplement of penicillin (5000 IU/mL), streptomycin (5 mg/mL), and 10% (*v*/*v*) FBS. The VUmc cell lines were cultured with Dulbecco’s Modified Eagle (DMEM) medium and were supplemented with 10% (*v*/*v*) FBS and antibiotics, with the exception of cell line KM12 which was cultured with RPMI 1640 medium with L-glutamine, 10% FCS, and antibiotics. The cells were kept in a cell incubator with 5% CO_2_ at 37 °C in humidified air. The cells were harvested when 80% confluence was reached and a trypsin/EDTA solution in 1× PBS was added to detach cells, followed by termination of the enzyme activity by a mixture of trypsin and a medium in a ratio of 2:5 (*v*/*v*). Cell counting was performed with a TC20 automated cell counter based on trypan blue staining. After washing the cells twice with 5 mL of 1× PBS, the cells were aliquoted to 2 × 10^6^ cells/mL of 1× PBS and centrifuged at 1500× *g* for 3 min. The cell pellets were stored at −20 °C.

### 4.3. N-Glycan Release

*N*-glycans were released from the cells as previously described with some slight modifications [72]. In brief, 96-well plates with hydrophobic Immobilon-P PVDF membrane were preconditioned with 200 µL 70% ethanol, and 3 × 200 µL mQ water. Simultaneously, to resuspend the cell pellets containing 2 × 10^6^ cells, 100 μL of lysis buffer were added, followed by 60 min sonication at 60 °C. In total, 25 µL of the cell lysate (containing 5 × 10^5^ cells) was applied to the preconditioned PVDF membrane wells. Protein denaturation was performed by adding 75 mL of a denaturation mixture, consisting of 72.5 µL 8 M GuHCl and 2.5 µL 200 mM DTT, to each well. Subsequently, the sample mixture was incubated in a humidified plastic box at 60 °C for 60 min. The plate was washed 3 times with mQ to remove the remaining denaturation agents via centrifugation at 500× *g* for approximately 2 min. The *N*-glycans were released from the denatured proteins by adding, subsequently, 2 µL of PNGase F (20 units) and 13 µL of mQ to each well followed by incubation at 37 °C for 15 min. Additionally, 15 µL of mQ water was added to each well and an overnight incubation was started in a humidified plastic box at 37 °C. Released *N*-glycans were collected by centrifugation at 500× *g* for 1 min and by washing the wells 3 times with mQ. The collected flow-through and washes were pooled for further processing. To hydrolyze the glycosylamine forms of the released *N*-glycans, 20 µL of 100 mM ammonium acetate (pH 5) was added to the collected *N*-glycans, followed by horizontal shaking for 10 min. Finally, the samples were dried in a SpeedVac concentrator.

### 4.4. N-Glycan Reduction and Purification

To reconstitute and reduce the released *N*-glycans [72], 40 µL of freshly prepared 1M NaBH_4_ in 50 mM KOH were added to each well. The samples were incubated in a humidified plastic box for 3 h at 50 °C. Afterward, the reaction was quenched by adding 4 µL of glacial acetic acid to each well, which also neutralized the sample. Samples were desalted in 96-well filter plates self-packed with cation exchange resin Dowex 50W X8. The packed plate was preconditioned by washing 3 times with 100 µL of 1 M HCl, followed by applying 3 times 100 µL of MeOH and 3 times 100 µL of mQ with centrifugation of 500 rpm for 1 min in between. The samples were applied onto the preconditioned plate and centrifuged at 500 rpm for 1 min followed by washing with two times 40 µL of mQ using centrifugation (2000 rpm for 3 min). The collected flow through and wash were combined and dried using a SpeedVac concentrator. To remove the remaining borate, 150 µL of MeOH were added 3 times to each well during the drying procedure. The cleanup of the samples was carried out on a 96-well filter plate packed with 60 µL (approximately 6 mg) of bulk sorbent carbograph slurry in MeOH which was preconditioned by applying 3 times 100 µL of 80% MeCN in mQ containing 0.1% TFA, subsequently followed by adding 3 times 100 µL mQ with 0.1% TFA. After sample loading (in 0.1% TFA), the columns were washed by applying 3 times 100 µL mQ with 0.1% TFA followed by elution adding 3 times 40 µL of 60% MeCN in mQ containing 0.1% TFA. The eluate was dried in a SpeedVac concentrator.

### 4.5. Analysis of N-Glycan Alditols on PGC Nano-LC-ESI-MS/MS

The purified *N*-glycan alditols were resuspended in 15 µL of mQ. A total of 5 µL of the sample was loaded onto a Hypercarb PGC trap column (5 µm Hypercarb Kappa, 320 µm × 30 mm, packed in the house) with 98% buffer A (10 mM ammonium bicarbonate) at 6 μL/min of loading flow. The *N*-glycans were separated on a Hypercarb PGC nanocolumn (3 µm Hypercarb Kappa, 100 µm × 100 mm, in house packed) with a multistep gradient of buffer B (60% MeCN in 10 mM ABC) of 2–9% in 1 min, followed by 9–49% in 80 min at a flow rate of 0.6 µL/min using the Dionex Ultimate 3000 nanoLC system. The column was washed with a 95% buffer B for 10 min. The column was held at a constant temperature of 35 °C. The separated *N*-glycans were detected by an amaZon speed ion trap MS with a capillary voltage set at 1000 V in negative mode, the dry gas temperature at 280 °C at a flow of 3 L/min and the nebulizer at 3 pounds per square inch (psi). The target mass was set at *m*/*z* 1200. MS spectra were acquired within a range of *m*/*z* 500-1850. MS and MS spectra were generated by collision-induced dissociation (CID) on the top 3 precursors with an isolation width of 3 Th. To enhance sensitivity, isopropanol was used as a dopant for the dopant-enriched nitrogen gas [73]. The *N*-glycan analysis workflow for the cells, prepared in a 96-well plate, is described in Appendix A.

### 4.6. Data Analysis

Identification of *N*-glycan structures was performed on the basis of accurate mass, retention time on the PGC column previously described diagnostic fragment ions (e.g., cross-ring fragments), and known biosynthetic pathways of *N*-glycans [26,68]. A single-letter code was used to refer to the monosaccharides: H for hexose, N for *N*-acetylhexosamine, F for fucose, and S for *N*-acetylneuraminic acid.

Data analysis was conducted with Bruker Compass DataAnalysis software (version 5.0). Briefly, extracted ion chromatograms were produced by extracting the theoretical mass of the first three isotopes of all observed (singly and doubly charged) species. The peak area under the curve was produced by integrating each peak with a signal to noise ratio ≥ 6 for all the technical and biological replicates. Relative quantification was calculated on the total area of all *N*-glycans detected in one sample normalized to 100%. “R” software (version 4.2.1) was used for further data analysis and visualization with packages “tidyverse”, “readxl”, “corrplot”, “Rcpm”, “pcaMethods”, “stringi”, “ggplot2”, “ggrepel”, “reshape2”, “ggpubr” and “tidyHeatmap”.

## Figures and Tables

**Figure 1 ijms-24-04842-f001:**
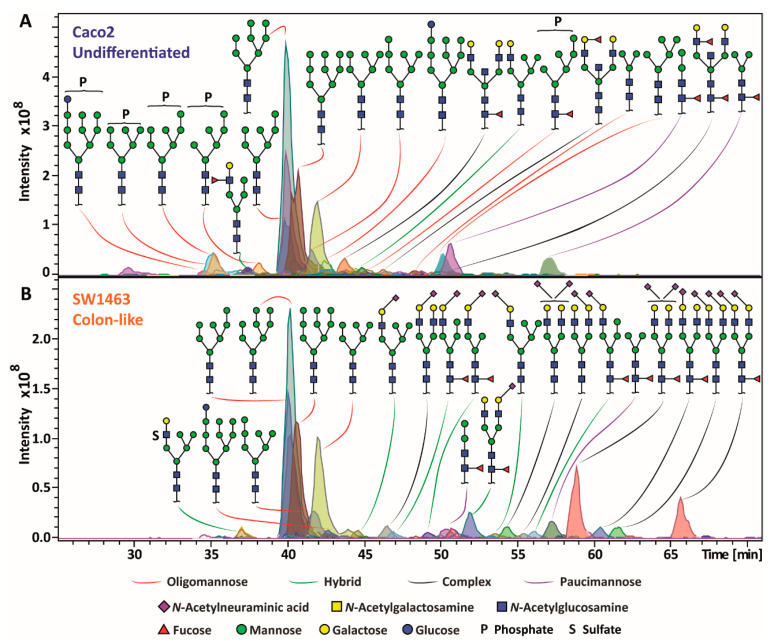
*N*-glycan profiles of two exemplary colorectal cancer (CRC) cell lines. (**A**) The Caco2 cell line was characterized by phosphorylated high mannose *N*-glycans as well as *N*-glycans carrying Lewis antigens. (**B**) Sialylated *N*-glycans were expressed by the SW1463 cell line. Sialic acid tilted to the left and right indicates α2-3 linkage and α2-6 linkage, respectively. Unidentified linkages are indicated with a sialic acid pointing vertically.

**Figure 2 ijms-24-04842-f002:**
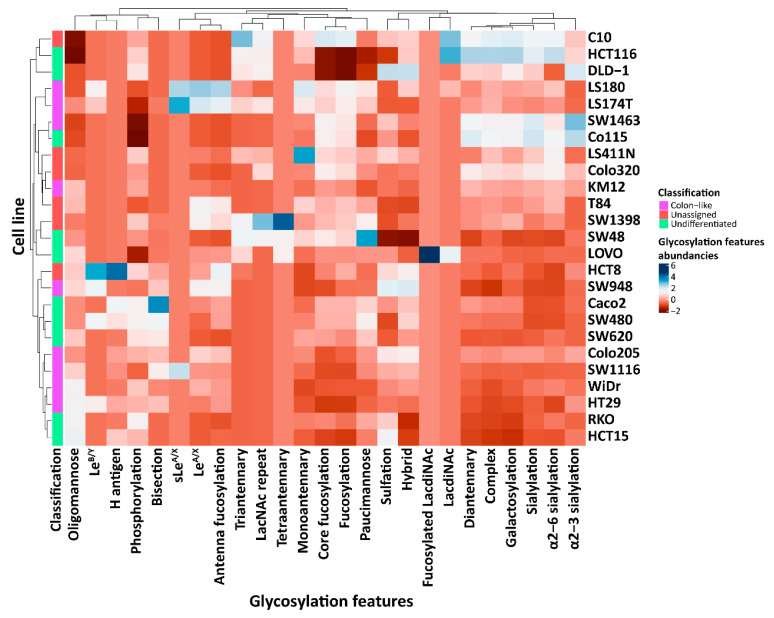
Distribution of *N*-glycosylation features in 25 CRC cell lines. The relative quantification of each glycosylation feature (bottom) was calculated for each cell line (right) and displayed in a clustered heatmap. The classification of the CRC cell lines is marked using color codes.

**Figure 3 ijms-24-04842-f003:**
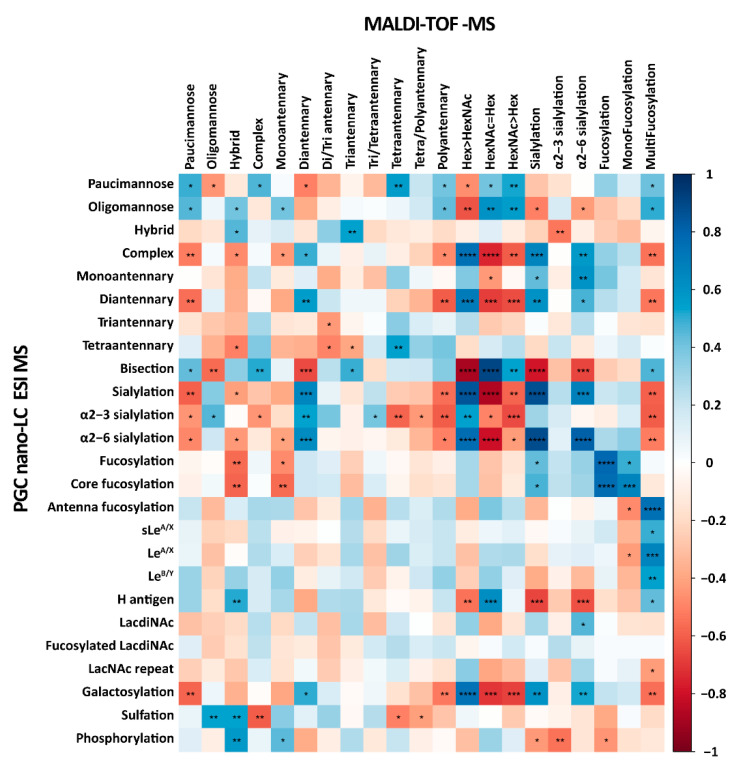
The correlation of *N*-glycan features in CRC cell lines measured by porous graphitized carbon nano-liquid chromatography electrospray ionization tandem mass spectrometry (PGC-nano-LC-ESI-MS) and matrix-assisted laser desorption/ionization time of flight-mass spectrometry (MALDI-TOF-MS). The Spearman correlation analysis was performed for the association of glycosylation features of *N*-glycans observed for the same set of CRC cell lines measured by PGC-nano-LC-ESI-MS in the present study and measured by MALDI-TOF-MS in the previous study [16]. Significant values are marked with * (*p* ≤ 0.05), ** (*p* ≤ 0.01), *** (*p* ≤ 0.001) and **** (*p* ≤ 0.0001). The color bar for correlation coefficient is displayed on the right.

**Figure 4 ijms-24-04842-f004:**
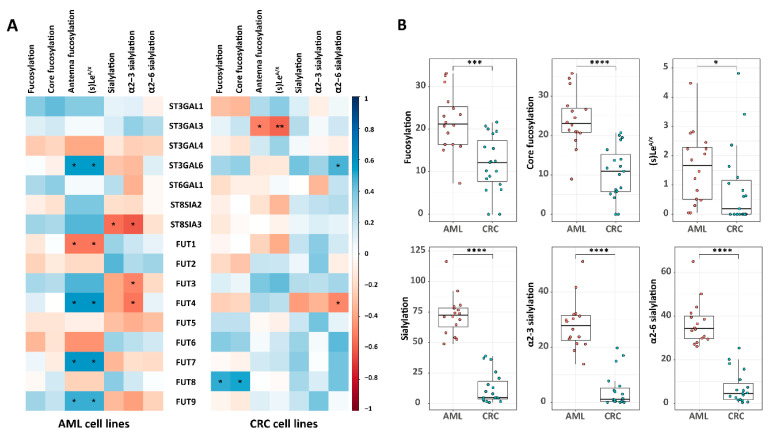
Correlations of glycosylation with relevant glycosyltransferases (GTs) and differences of glycosylation features in CRC and acute myeloid leukemia (AML) cell lines. (**A**) Correlation analysis was conducted between glycosylation features (relative quantification %) and corresponding GTs in CRC cell lines with the Spearman method. (**B**) The differences between groups (AML vs. CRC cell lines) were tested with Wilcoxon–Mann–Whitney nonparametric statistical test. The glycosylation features with significant differences between AML and CRC cell lines were displayed in (**B**). *p*-values were corrected by the Benjamini–Hochberg method. Significant values are marked with * (*p* ≤ 0.05), ** (*p* ≤ 0.01), *** (*p* ≤ 0.001), and **** (*p* ≤ 0.0001). The R2 scale is indicated in the right key bar of subfigure (**A**) (blue: positive correlation; red: negative correlation).

**Figure 5 ijms-24-04842-f005:**
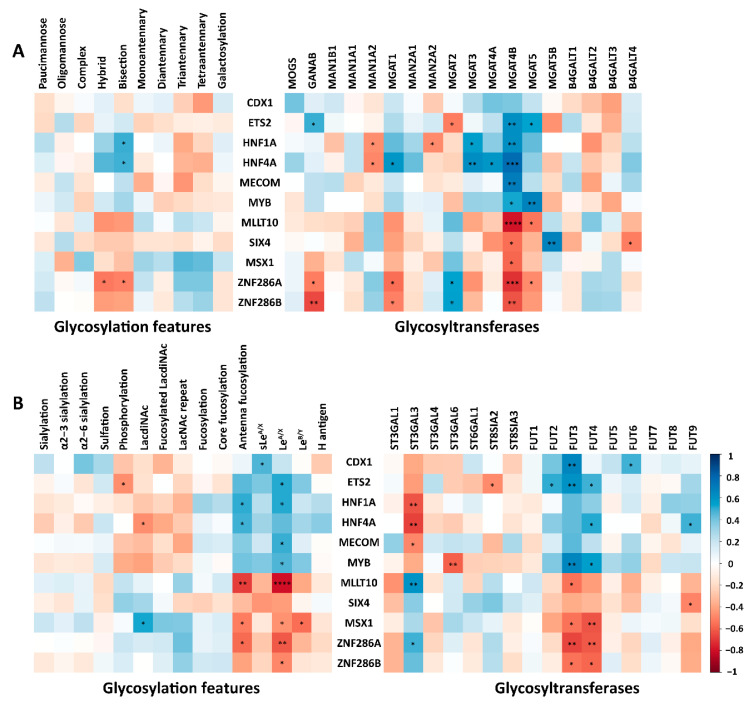
(**A**,**B**) Associations between transcription factors (TFs) and glycosylation features (**left** panel) and GTs (**right** panel). The association between abundances of glycosylation features and mRNA expression levels of relevant genes taken from a publicly available dataset [15] was tested using Spearman correlation. Significant values are marked with * (*p* ≤ 0.05), ** (*p* ≤ 0.01), *** (*p* ≤ 0.001), and **** (*p* ≤ 0.0001). The correlation coefficient is displayed on the right key bar (blue: positive correlation; red: negative correlation).

## Data Availability

The raw mass spectrometric data files supporting the findings of this study are available in GlycoPOST in mzXML format, with the identifier GPST000302.

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
