# Peer review of "In-Depth Analysis of the N-Glycome of Colorectal Cancer Cell Lines"

_ijms, 2023, doi:10.3390/ijms24054842_

Round 1

Reviewer 1 Report

The manuscript reported by Langeveen-Kammeijer described detailed N-glycome analysis of 25 colorectal cancer cell lines based on a modern state-of-art LC-MS technique. They elucidated 139 glycan structures of which abundance together with their structures were used for statistical analysis. In the end, they found out the importance of a transcription factor, CDX1 that correlated with a fucosyltransferase that further related with some marker motif carbohydrate antigens.

Overall investigation is outstanding and the manuscript is well written. Also, this reviewer appreciate the "right direction" of the research.

The followings are some comments.

Line 48: Meaning of retention of tissue molecular features is not clear.

Lines 144, 150 and others: There are different ways of describing drying process such as drying using a vacuum centrifuge and SpeedVac concentrator. Are they same?

N-Glycan structures: A reviewer could not find how they elucidated all the glycan structures. Since this is the basis of the entire investigations, it should be clarified in the experimental section.

In the experimental setup, the authors used a sum of first three isotopes of singly and doubly charged ions. It might be just a way of expression but the term abundance may not be correct word because ionization potentials of individual ions are different and so the peak area does not reflect the abundance of those molecules. It is okay for specialists who understands the basics of MS but may not give accurate information to broad readers.

The "abundance" obtained from single analysis of LC-MS are analyzed statistically. It may be a common sense but describing two decimals seems to me too much to deal with. Let's say, if you do the same experiment twice,  you notice few percent would be negligible and provably that could be a reason of handling data from single run.

Line 189 and others: The abbreviations of glycan structures such as H4N5S1 is not common and impossible to follow by non-specialists. The excel file available as a supporting information is essential for readings. A reviewer suggest to find out a better way to either express or to guide readers to SI.

Lines 216–231: There is only one figure 2.

Line 233: The "(fucosylated) LacdiNAc..." It is not clear what fucosylations.

Line 251: Sd<upper script>a</upper script>

Line 258: (s) LeA/X is confusing. It may be +/-sLeA/X.

Line 269: D–221 is not clear.

Line 289: A sentence ...was supported by Supplementary Information, Fig. S6... has to be described in text.

Line 343: delete redundant "tumor-associated carbohydate antigens"

Line 358: ...it lacke the ability to separate ... is not entirely true. Some can still carry out such things based on an off-line LC-MALDI MS setups.

Line 399: Is ST6GAL1 gene?  If that is so, what the cleavage of the gene mean.

Author Response

Response to Reviewer’s Comments

Please find below a point-by-point response to the valuable comments and issues raised by the reviewers:

Reviewer 1

Comments and Suggestions for Authors

  • The manuscript reported by Lageveen-Kammeijer described detailed N-glycome analysis of 25 colorectal cancer cell lines based on a modern state-of-art LC-MS technique. They elucidated 139 glycan structures of which abundance together with their structures were used for statistical analysis. In the end, they found out the importance of a transcription factor, CDX1 that correlated with a fucosyltransferase that further related with some marker motif carbohydrate antigens.
  • Overall investigation is outstanding and the manuscript is well written. Also, this reviewer appreciate the "right direction" of the research.

Response: We would like to thank the reviewer for their valuable time and positive evaluation of our study. Please find below our responses to your feedback.

The followings are some comments:

  1. Line 48: Meaning of retention of tissue molecular features is not clear.

Response: We thank the reviewer for pointing this out. We have revised the text accordingly. (See lines 48-49 of the manuscript with track change)

Lines 48-49:“…retain tissue molecular features with regard to DNA, RNA and proteins…”

  1. Lines 144, 150 and others: There are different ways of describing drying process such as drying using a vacuum centrifuge and SpeedVac concentrator. Are they same?

Response: The remark is correct and these two procedures are indeed the same. To avoid confusion, we have changed it to “SpeedVac concentrator” throughout the manuscript.

  1. N-Glycan structures: A reviewer could not find how they elucidated all the glycan structures. Since this is the basis of the entire investigations, it should be clarified in the experimental section.

Response: We would like to thank the reviewer for mentioning this. We have added the description of the N-glycan assignment in the Data analysis section (See lines 167-169 of the manuscript with track change).

Lines 167-169: “Identification of N-glycan structures was performed on the basis of accurate mass, retention time on the PGC column, previously described diagnostic fragment ions (e.g. cross-ring fragments) and known biosynthetic pathways of N-glycans.28, 29

  1. In the experimental setup, the authors used a sum of first three isotopes of singly and doubly charged ions. It might be just a way of expression but the term abundance may not be correct word because ionization potentials of individual ions are different and so the peak area does not reflect the abundance of those molecules. It is okay for specialists who understands the basics of MS but may not give accurate information to broad readers.

Response: In order to indicate the limitation mentioned by the reviewer, we included a sentence in the Results section. (See lines 185-188 of the manuscript with track change)

Lines 185-188: “Relative abundances of the N-glycans based on the first three isotopic peaks of singly and doubly charged ions were determined, which may somewhat deviate from the actual relative abundances due to possible biases in sample preparation and mass spectrometric detection.”

  1. The "abundance" obtained from single analysis of LC-MS are analyzed statistically. It may be a common sense but describing two decimals seems to me too much to deal with. Let's say, if you do the same experiment twice, you notice few percent would be negligible and provably that could be a reason of handling data from single run.

Response: Thank you for your comments, the number of decimals has been reduced to one. (See lines 196-237 of the manuscript with track change)

  1. Line 189 and others: The abbreviations of glycan structures such as H4N5S1 is not common and impossible to follow by non-specialists. The excel file available as a supporting information is essential for readings. A reviewer suggest to find out a better way to either express or to guide readers to SI.

Response: We have added more information in the manuscript about the abbreviations that are used for the monosaccharides. (See lines 169-170 and 200-201 of the manuscript with track change)

Lines 169-170: “A single-letter code was used to refer to the monosaccharides: H for hexose, N for N-acetylhexosamine, F for fucose and S for N-acetylneuraminic acid”

Lines 200-201: “Structural information of the identified N-glycans can be found in Supplementary Information, Table S1.”

  1. Lines 216–231: There is only one figure 2.

Response: We would like to thank the reviewer for pointing out the discrepancy. The manuscript has been corrected accordingly.

  1. Line 233: The "(fucosylated) LacdiNAc..." It is not clear what fucosylations.

Response: We have provided a clarification of the expected fucosylation site in the manuscript. (See lines 240-241 of the manuscript with track change)

Lines 240-241: “N-glycans carrying LacdiNAc with or without fucose linked to GlcNAc via α1-3 linkage…”

  1. Line 251: Sd<upper script>a</upper script>

Response: We have adjusted the format throughout the manuscript based on the reviewer’s feedback.

  1. Line 258: (s) LeA/X is confusing. It may be +/-sLeA/X.

Response: To avoid confusion, the (s)LeA/X has been written separated as LeA/X and sLeA/X throughout the entire manuscript.

  1. Line 269: D-221 is not clear.

Response: We have added more information to explain the D-221 ion. (See lines 277-278 of the manuscript with track change)

Lines 277-278: “…the presence of D-221 ion, which is formed from the additional loss of the β1-4 linked GlcNAc from the D ion of bisected N-glycans,…”

  1. Line 289: A sentence ...was supported by Supplementary Information, Fig. S6... has to be described in text.

Response: Thank you for the suggestion. More explanation about Supplementary Information, Fig. S6 has been added. (See lines 298-300 of the manuscript with track change)

Lines 298-300: “An overall higher expression of LeA/X and sLeA/X was found for AML cell lines in comparison to CRC cell lines (Figure 4B), with accordingly higher expression of the FUT4/7 in AML cell lines compared to CRC cell lines (Supplementary Information, Fig. S7). ”

  1. Line 343: delete redundant "tumor-associated carbohydate antigens"

Response: The “tumor-associated carbohydate antigens” in lines 354-355 have been removed.

  1. Line 358: ...it lacks the ability to separate ... is not entirely true. Some can still carry out such things based on an off-line LC-MALDI MS setups.

Response: While the reviewer is correct that using an off-line LC-MALDI-MS platform would allow to elucidation of isomers a standalone MALDI-MS platform is not. We have adjusted the sentence to clarify this. (See lines 368-370 of the manuscript with track change)

Lines 368-370: “However, while MALDI-TOF-MS is efficient for high-throughput glycan profiling, it lacks the ability to separate glycan isomers or provide structural information when no separation is applied prior to the measurement.28

  1. Line 399: Is ST6GAL1 gene? If that is so, what the cleavage of the gene mean.

Response: We thank the reviewer for their question. With cleavage of the gene we meant proteolytic cleavage of the enzyme which is encoded by the gene. We have revised the manuscript accordingly. (See lines 411-412 of the manuscript with track change)

Lines 411-412: “…is responsible for the proteolytic cleavage of beta-galactoside alpha-2,6-sialyltransferase 1 enzyme (encoded by ST6GAL1).70

Reviewer 2 Report

The study analyzed the N-glycosylation of 25 colorectal cancer cell lines using mass spectrometry, finding 139 N-glycans with high diversity. It also linked CDX1, (s)Le antigen, and glycosyltransferases FUT3/6. This study contributes to finding new glyco-biomarkers for colorectal cancer.

The manuscript is easy to read, well-written and well-worth publishing. I strongly support its publication.

Author Response

Response to Reviewer’s Comments

Reviewer 2

Comments and Suggestions for Authors

  • The study analyzed the N-glycosylation of 25 colorectal cancer cell lines using mass spectrometry, finding 139 N-glycans with high diversity. It also linked CDX1, (s)Le antigen, and glycosyltransferases FUT3/6. This study contributes to finding new glyco-biomarkers for colorectal cancer.
  • The manuscript is easy to read, well-written and well-worth publishing. I strongly support its publication.

Response: We would like to thank the reviewer for their positive feedback.

Reviewer 3 Report

In this fundamental study, the authors presented the value of
analyzing the depth analysis of N-glycosylation of CRC in diagnostics for CRC. The author's newly discovered profound N-glycomic diversity among
the studied CRC cell lines, with the elucidation of some 139 N-glycans
CDX1 and CDX1 contribute to the expression of (s)Le antigen by regulating FUT3/6. It is a well-written article. I humbly request  the authors provide the flow diagram of the experimental method in the figure.

Author Response

Response to Reviewer’s Comments

Please find below a point-by-point response to the valuable comments and issues raised by the reviewers:

Reviewer 3

Response: We thank the reviewer for their positive feedback in regard to our manuscript.

Comments and Suggestions for Authors

  • In this fundamental study, the authors presented the value of analyzing the depth analysis of N-glycosylation of CRC in diagnostics for CRC. The author's newly discovered profound N-glycomic diversity among the studied CRC cell lines, with the elucidation of some 139 N-glycansCDX1 and CDX1 contribute to the expression of (s)Le antigen by regulating FUT3/6. It is a well-written article. I humbly request the authors provide the flow diagram of the experimental method in the figure.

Response: We thank the reviewer for the comment. We have added workflow as Supplementary Information Figure 1. (See lines 164-165 of the manuscript with track change)

Lines 164-165: “The N-glycan analysis workflow for cells, prepared in a 96-well plate, is described in Supplementary Information, Fig. S1.”

Reviewer 4 Report

The paper describes the glycomics analysis in various colorectal cancer cell lines.

The study has a lot of cell lines and the experimental data of glycome analysis and also enzymes.

The paper has high impact for cancer glycomics.

I have only a few comments.

1) I had a difficulty for understand that how did authors separate the structural isomers (ex; glucose and galactose).

2) Is it able to establish the multiple biomarkers set for differentiations of various cancer types from this experimental data? 

Author Response

Response to Reviewer’s Comments

Please find below a point-by-point response to the valuable comments and issues raised by the reviewers:

Reviewer 4

Comments and Suggestions for Authors

The paper describes the glycomics analysis in various colorectal cancer cell lines. The study has a lot of cell lines and the experimental data of glycome analysis and also enzymes. The paper has high impact for cancer glycomics.

Response: We thank the reviewer for their valuable time reading our manuscript. We have addressed the points raised by the reviewer as below.

Comments:

  1. I had a difficulty for understand that how did authors separate the structural isomers (ex; glucose and galactose).

Response: Thank you for the comment. The reviewer is correct that structural elucidation of hexoses could be rather difficult, however, in order to identify the structures of isomeric N-glycans we used the following information:

  • Diagnostic ions observed in the MS/MS spectra of N-glycans produced in the negative ionization mode (e.g. cross-ring fragments as per Harvey et.al (https://doi.org/10.1007/s13361-018-1950-x))
  • Identified N-glycans that were described in a previous study1 ;
  • The relative retention time of glycans on PGC was used as well to confirm the structure of glycans; to exemplify, as oligomannosidic type of N-glycans eluted earlier than complex N-glycans.
  • The previously well-described in literature N-glycan biosynthetic pathways also help with the assignment of N-glycans.

This information has been added to the manuscript(See lines 167-169 of the manuscript with track change)

Lines 167-169: “Identification of N-glycan structures was performed on the basis of accurate mass, retention time on the PGC column, previously described diagnostic fragment ions (e.g. cross-ring fragments) and known biosynthetic pathways of N-glycans.28, 29

  1. Is it able to establish the multiple biomarkers set for differentiations of various cancer types from this experimental data?

Response: The method that was engaged in this study can be applied to other samples (e.g. cell lines, serum or tissue samples derived from other studies) with possible adjustments of the preceding sample preparation steps. Therefore, we indeed believe that it will be beneficial to investigate the glycomic profiles of other cancer types to gain a fuller understanding of their molecular signatures as well as to discover possible glycan-related biomarkers. In addition, in this study, we made the first step towards the exploration whether the glycosylation differs between various types of cancer by investigating the glycomic and transcriptomic features of AML and CRC. Here distinct features were found per cancer type. However, further studies are needed to probe the similarities and differences between a wide range of cancer types. Hopefully, this will result in establishing biomarker panels aiding in the differentiation of various cancer types.

Reference:

  1. Everest-Dass, A. V.; Abrahams, J. L.;  Kolarich, D.;  Packer, N. H.; Campbell, M. P., Structural feature ions for distinguishing N- and O-linked glycan isomers by LC-ESI-IT MS/MS. J Am Soc Mass Spectrom 2013, 24 (6), 895-906.

Round 2

Reviewer 4 Report

The paper describes the glycomics analysis in various colorectal cancer cell lines.

The response comments and the revised paper were checked and I've understood them.

I have no more comments for improvement of it.